# Predicting intentions towards long-term antidepressant use in the management of people with depression in primary care: A longitudinal survey study

**Rachel Dewar-Haggart**[1]*, **Ingrid Muller**[1], **Felicity Bishop**[2], **Adam W. A. Geraghty**[1], **Beth Stuart**[3], **Tony Kendrick**[1]

**1** Primary Care Research Centre, University of Southampton, Southampton, United Kingdom, **2** Centre for Clinical and Community Applications of Health Psychology, University of Southampton, Southampton, United Kingdom, **3** Wolfson Institute of Population Health, Queen Mary University of London, London, United Kingdom

☉ These authors contributed equally to this work.

* rr1d12@soton.ac.uk

## Abstract

### Background

Over the last two decades, antidepressant prescribing in the UK has increased considerably, due to an increased number of people staying on antidepressants for longer. Even when treatment is no longer clinically indicated, qualitative research suggests many people continue due to a fear of depressive relapse or antidepressant withdrawal symptoms. The quantitative effects of peoples' beliefs and attitudes towards long-term antidepressant use remain relatively unexplored.

### Objectives

To determine the extent to which beliefs and attitudes towards antidepressant treatment are associated with intentions to stop or continue long-term use; and whether intentions translate into actual discontinuation.

### Methods

A questionnaire survey formed the main component of an embedded mixed-methods study. Twenty general practices posted questionnaires to adults aged over 18 receiving continuous antidepressant prescriptions for over two years. Outcomes and explanatory variables were determined using an extended model of the Theory of Planned Behaviour, conducting exploratory descriptive and regression analyses. The primary outcome was participants' intentions to discontinue antidepressants. The secondary outcome of behaviour change was determined by any change in antidepressant dosage at six months.

provided the original author and source are credited.

**Data availability statement:** The Ethical Approvals provided by the Sponsor (University of Southampton), NHS Research Ethics Committee, and Health Research Authority, states that "Access to raw data and right to publish freely by all investigators in study or by Independent Steering Committee on behalf of all investigators" will not form part of the dissemination plan. Furthermore, participants have not provided consent for the data to be shared anonymously for other ethically approved research in the future. Any person who makes a request to the University for information not made available through the publication scheme is entitled (subject to the exemptions enumerated in the Act) to be informed in writing whether the University holds the information requested and if so, to have the information communicated to him or her. Please request information by emailing foi@soton.ac.uk.

**Funding:** RDH received funding to complete a PhD. This study/project is funded by the National Institute for Health and Care Research (NIHR) School for Primary Care Research (project reference SPCR-097). The views expressed are those of the author(s) and not necessarily those of the NIHR or the Department of Health and Social Care. URL: https://www.spcr.nihr.ac.uk/ The funders did not play any role in the study design, data collection and analysis, decision to publish or preparation of the manuscript.

**Competing interests:** I have read the journal's policy and the authors of this manuscript have the following competing interests: TK, AWG and BS have received grant funding from the NIHR Research for Patient Benefit programme to conduct research into helping people come off inappropriate long-term antidepressants.

## Results

277 people were surveyed from 20 practices, with 10 years median antidepressant duration. Mean questionnaire scores for intention and subjective norms towards starting to come off antidepressants were low, and 85% of participants declared that continuing their antidepressant was necessary. Prescribing outcomes retrieved from 175 participants' medical records six months after they completed the survey found 86% had not changed their antidepressant, 9% reduced the dose, only 1% discontinued their antidepressant, and 4% increased the dose. All Theory of Planned Behaviour constructs and concerns were associated with intentions, with more favourable attitudes towards stopping and subjective norms having the strongest associations towards intentions to discontinue antidepressant use.

## Conclusion

Given few intentions to stop taking antidepressants, patients should be made more aware of the importance of ongoing antidepressant monitoring and review from their primary care practitioners. This would promote discussion to support an attitudinal change and initiation of antidepressant tapering where appropriate.

## Introduction

Over the past two decades, antidepressant prescribing rates have risen considerably, nearly doubling between 2008 and 2018 [1,2]. Between 2015 and 2018, the rate of antidepressant prescribing in primary care increased from 15.8% to 16.6% [3,4]; with 7.3 million people prescribed antidepressants in 2017/18, at an annual cost of approximately £266 million [5]. The considerable rise in the volume of antidepressant prescribing in primary care is due to an increased number of people receiving continuous antidepressant treatment for longer [2,6–11].

A third to a half of people taking long-term antidepressants may have no evidence-based indications to continue treatment, and could try to stop [12]. Long-term outcomes of antidepressant-treated depression are generally poor [13], and antidepressants may additionally pose the risk of adverse long-term iatrogenic effects such as sexual problems, weight gain, feeling emotionally numb and the perception of being addicted to medication [14–17]. In people over 65, adverse effects associated with antidepressant use include falls, seizures, strokes, low blood sodium, and cardiac arrhythmias [18].

The NICE guidelines emphasise the need for continued monitoring and review of people on long-term antidepressant treatment [19]. However, few review consultations happen with people who use antidepressants long-term, with the percentage of people reviewed during each year of treatment decreasing over ten years [20,21]. Reviewing long-term antidepressant use can reduce drug burden, with a primary care pharmacist-led study showing that around 15% of people who had an active review had their antidepressant therapy altered, which led to a reduction in antidepressant prescribing [22]. This emphasises the importance of GPs to invite people who have been on antidepressants for more than two years to a review [23]. However, minimising inappropriate long-term antidepressant use can be challenging for GPs [24–27], due to perceived patient demand for antidepressants treatment and a lack of opportunity to have review consultations [28]. Furthermore, people may prefer requesting repeat prescriptions remotely or are ambivalent about the need for a consultation if the GP continues to approve prescriptions without a review [29,30].

People with a stronger belief in the effectiveness of medication are more likely to be taking antidepressants, more likely to believe that their condition has a chronic timeline, and more likely to be currently depressed [31–33] Individuals have a greater perceived need for antidepressants if they believe their depression is caused by chemical imbalances or is hereditary [34,35] These findings suggest that a greater belief in the chronic and biochemical nature of depression will lead to longer-term antidepressant treatment, as patients may believe that pharmacological interventions are more effective at symptom management than non-drug treatments. However, higher self-efficacy in managing depressive symptoms and a belief in using talking therapies or engaging in activities such as exercise or keeping busy to manage depression is associated with improved depression outcomes [31,36]

While there is considerable qualitative evidence on the issues surrounding ongoing antidepressant treatment in primary care, the quantitative effects of peoples' beliefs and attitudes towards long-term antidepressant use remain relatively unexplored.

The aims of this study were to investigate the extent to which beliefs and attitudes towards depression and antidepressant treatment are associated with intentions to stop or continue long-term antidepressant use; and whether these intentions translate into actual behaviour of antidepressant discontinuation. A further objective was to determine how well participants' beliefs and attitudes can be explained by the Theory of Planned Behaviour (TPB) [37].

## Materials and methods

### Study design

A longitudinal, mixed-methods design using a quantitative questionnaire survey study with nested qualitative interview study was used, with data collection occurring concomitantly [38, 39] The findings from the qualitative study will be reported elsewhere. A copy of the study protocol is provided as S1 File.

### Setting

Twenty group general practices from the Clinical Research Network (CRN): Wessex and CRN: West of England were recruited to the study from November 2017. Participant recruitment began in February 2018 and ended in February 2019.

### Participants

Practices were asked to conduct a database search to identify patients over the age of 18 who had been continuously receiving antidepressant prescriptions for two years or longer. The duration of two years or longer was defined based on the National Institute for Health and Care Excellence (NICE) guidance [19] that individuals should remain on antidepressants for at least two years if they have had two or more depressive episodes in the recent past or are at risk of relapse; but could decide with their GP whether to stop or continue treatment at this time. Practices were given both a list of British National Formulary (BNF) [40] antidepressant names and Read codes [41] for diagnoses and symptoms of depression to conduct the search. Patients were excluded if they were prescribed antidepressants for conditions other than depression, had a comorbid psychiatric condition or depression managed in secondary care, or were terminally ill, lacking capacity, or deemed unable to take part after screening by a GP. Participants were not excluded based on their severity of depression or if they had any comorbid physical conditions.

Eligible participants were sent an invitation pack in the post by their GP practice. Each practice was asked to send packs to up to 140 patients. The pack included an information

sheet, questionnaire booklet, and consent form. The study was ethically approved through proportionate review by Yorkshire & The Humber – Leeds East Research Ethics Committee (REC ID: 17/YH/0223). Completion of the questionnaire indicated implied consent [42]. However, participants were required to provide written consent for their GP practice to provide anonymised medical notes data. The authors did not have direct access to participants' medical records, and only data that were relevant for the outcome of the study (antidepressant prescribing data and/or any record of consulting with a primary care practitioner for a mental health review within six months of the participant completing the questionnaire) were obtained. Requests for this data from the GP practices were made up to March 2020, due to the Covid-19 pandemic affecting practices' capacity to continue research.

## Outcomes and variables

Outcome and explanatory variables were determined by using an extended model of the (TPB) (Fig 1) [37]. The primary outcome was participants' declared intentions to start to come off antidepressants. The secondary outcome of actual behaviour change was determined by a reduction in antidepressant prescription dosage or attending an appointment to discuss possible discontinuation within the six-months of completing the questionnaire. In addition to the TPB constructs of attitude, subjective norms, and perceived behavioural control (PBC), further constructs of global beliefs [33,43–47], past behaviour [43,48], symptom severity [19], and current antidepressant duration [33,45] were hypothesised to be associated with intentions to stop or continue long-term antidepressant use and added to the model [37,43,49–51]. Global beliefs were defined as beliefs about depression and antidepressant use. As the TPB suggests that underlying global beliefs may determine attitudes towards a behaviour [43,44], the association between global beliefs and attitudes towards starting to come off antidepressants in the next six months was also investigated.

## Data measurement

The questionnaire survey participants were asked to complete is included as S2 File. The medical notes data form that practices were asked to complete is included as S3 File.

## Demographic characteristics

Participants provided demographic characteristics including gender, age, ethnicity, marital status, number of dependants, level of education, and occupation.

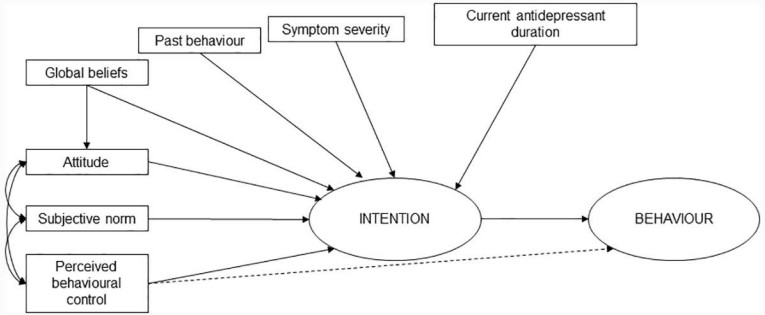

**Fig 1. An extended model of the Theory of Planned Behaviour.**

## Intention, attitudes, subjective norms, and perceived behavioural control

Intention, attitudes, subjective norms and perceived behavioural control (PBC) towards starting to come off antidepressant treatment were measured using Likert rating-scale items, created using guidance for developing questionnaire items based on the TPB [37,52]. Items were developed and refined through cognitive interviews as part of RDH's PhD [53]. Intention was measured using the 'generalised intention' method, where participants were asked about their intentions around their own health-related behaviour [52]. Attitudes were measured by using both experiential items (i.e., feelings about starting to come off antidepressants in the next six months) and instrumental items (i.e., beliefs about whether starting to come off antidepressants in the next six months will achieve a positive or negative outcome) [52]. For subjective norms, items measuring injunctive normative beliefs were used to determine whether participants agreed with statements that particular individuals or groups felt they should start to come off antidepressants in the next six months [54]. PBC was measured using items focussing on patients' perceived self-efficacy and controllability in starting to come off antidepressants in the next six months [52] The number of items included within each variable is shown in Table 1. Items with Cronbach's alpha correlations of α > .60 suggested acceptable internal consistency [52,55]. A mean score for the items within each of the items relating to intention, attitude, subjective norm, and PBC was calculated, with higher scores indicating stronger beliefs. Additional descriptive statistics around attitudes towards antidepressant discontinuation were measured using items from the Patient Attitudes towards Deprescribing (PATD) Questionnaire [56]. Items asked participants to indicate their agreement with statements on a 5-point Likert scale (Strongly Disagree, Disagree, Uncertain, Agree, Strongly Agree). Frequency data for each item were calculated as no scoring system was created for the PATD Questionnaire [56].

**Table 1. Explanatory variables included in the extended model of the Theory of Planned Behaviour.**

| Variable | Description | Survey Measure | Items |
|---|---|---|---|
| Intention | Intentions towards starting to come off antidepressants in the next six months | TPB | 3 |
| Attitude | Attitudes around the benefits and disadvantages of starting to come off antidepressants in the next six months | TPB | 7 |
| Subjective norms | Perceived opinions of others about starting to come off antidepressants in the next six months | TPB | 4 |
| PBC | Perceived self-efficacy and controllability around starting to come off antidepressants in the next six months | TPB | 3 |
| Global Beliefs | | | |
| *Necessity* | Beliefs around the necessity of antidepressants | BMQ-Specific | 5 |
| *Concerns* | Concerns about taking antidepressants | BMQ-Specific | 5 |
| *Medication* | Beliefs around the need for antidepressants to control/cure depression | BDQ | 1 |
| *Physical* | Beliefs around the physical causes of depression (genetics, illness, chemical imbalance) | BDQ | 3 |
| *Chronic* | Beliefs around the chronic timeline of depression | BDQ | 2 |
| Past Behaviour | | | |
| *With doctor* | Stopping previous antidepressant treatment with a doctor's knowledge | PATD Questionnaire | 1 |
| *Without doctor* | Stopping previous antidepressant treatment without a doctor's knowledge | PATD Questionnaire | 1 |
| *Successfully stopped* | Duration of previous symptom free episode while off antidepressant treatment | Bespoke item | 1 |
| Current antidepressant duration | Current duration of antidepressant treatment | Bespoke item | 1 |
| Symptom severity | Current depression symptom severity | PHQ-8 | 8 |

## Global beliefs

Necessity and concern beliefs about antidepressants were measured using the Beliefs about Medicines-Specific Questionnaire (BMQ-Specific) [57]. Items were modified by changing the word '*medicines*' to '*antidepressants*' [47]. A total score of both *necessity* and *concern* items were calculated and interpreted as continuous scales, with higher scores indicating stronger beliefs in the necessity of, or greater concerns about taking antidepressants. *Physical cause*, *chronic timeline,* and *medication to control/cure* variables of the Beliefs about Depression Questionnaire (BDQ) [57]. were calculated by mean scores from 6-point Likert rating scales, where higher scores indicated stronger beliefs.

## Past behaviour, current treatment duration and symptom severity

Participants provided dichotomous data for past behaviour items adapted from the PATD Questionnaire [56], that asked whether participants had tried to stop antidepressants in the past either with, or without, their doctor's knowledge. A further measure of past behaviour was a bespoke dichotomous item asking whether participants had successfully stopped taking antidepressants. 'Successfully' was defined to participants as 'symptom free episode(s) while off antidepressant treatment'. Participants provided self-report data on the current duration of their antidepressant treatment in years and months. The Patient Health Questionnaire (PHQ-8) [58] was included to measure current symptom severity of depression. The PHQ-8 includes the same items as the PHQ-9 [59], but excludes question nine, which assesses thoughts of harm or suicidal ideas. The omission of this item only has a small effect on scoring, and identical thresholds are used for both the PHQ-8 and PHQ-9 questionnaires [58,60]. Symptom scores are categorised to five levels of severity: minimal (0–4), mild (5–9), moderate (10–14), moderately severe (15–19), and severe (20–24).

## Behaviour

GPs at participating practices conducted medical record reviews for the six-month period following participants' completion of the questionnaires. The reviews were conducted to measure the proportion of participants who consulted with a health professional at their GP surgery to review their mental health and determine whether they had started to discontinue treatment, indicated by a reduction in their prescribed antidepressant dosage. Participants were categorised into two groups based on their prescribing data: reduced (reduced or stopped) and did not reduce (increased, no change, or changed antidepressant type).

A summary of all explanatory variables included in the model, along with the number of questionnaire items used to measure each explanatory variable, are described in Table 1.

## Study size

Guidance on questionnaires using constructs from the TPB suggest a sample size of 80 participants, if a moderate effect size of 0.3 is expected following multiple regression analysis [52,61]. A 'rule of thumb' sample size estimate was calculated based on Green's procedure; accounting for the potential of a small effect size and potential of overfitting [62,63]. Approximately 405 participants would be required for a multiple regression analysis and was deemed feasible to obtain in a primary care setting, assuming a 10% response rate.

## Statistical methods

Data were analysed using SPSS version 26 [64]. Frequency distributions and means were calculated for participant characteristics and information on participants' antidepressant use and

history of depression. Pearson's correlation was conducted to determine whether there was an association between beliefs around the necessity of and concerns around antidepressant treatment.

Tests to ensure the data met the assumptions for multivariable analysis found heteroscedasticity of residuals; therefore parameter estimates with robust standard errors were calculated to account for this heteroscedasticity. Multiple linear regression with robust standard errors was conducted to determine whether the independent variables within the construct of 'global beliefs' had a linear relationship with attitudes towards antidepressant discontinuation and the extent of these relationships [63]. Sequential regression analysis was conducted to determine how well the constructs of the TPB model could explain intentions to start to come off antidepressants (Step 1), followed by the addition of global beliefs (Step 2), past behaviour (Step 3), current symptom severity (Step 4), and current antidepressant use (Step 5). These constructs were added to determine whether they could explain intentions to stop antidepressants over and above constructs from the TPB. Each construct was added to the regression analysis based on RDH's assumptions around the theoretical importance of each variable. A binomial logistic regression was anticipated to determine whether intentions and PBC towards starting to come off antidepressants predicted behaviour (indicated by a reduction in antidepressant dose) and to ascertain the effect of intentions to start to come off antidepressants on whether participants had at least one appointment with a health professional. However, nine of the 12 participants who reduced their antidepressants had studentized residuals ± 2.5. Examining the data suggested they were outliers as they all had low intention scores yet reduced their antidepressants six months after completing the questionnaire. Therefore parametric assumptions to conduct a logistic regression were not met. Data analysis was conducted using complete cases.

## Results

### Participants

Recruitment of participants is shown in Fig 2. Most patients approached were female (n = 1288, 70.9%), with a mean age of 55.5 years (SD = 15.3, range = 20–96). Three hundred and ninety-seven responses were received (16.9% of those approached). Questionnaires from 120 respondents were excluded from the study, with 68 excluded based on the self-report item for current antidepressant duration. Forty respondents reported antidepressant treatment duration of less than two years, 13 did not provide any information, and 15 provided data that were unclear, for example: "*don't know*", "*can't remember*", or "*years*". One person returned the questionnaire but later requested to withdraw from the study, including their questionnaire data, with no reason given. Two hundred and seventy-seven participants (11.8% of those approached) were entered into the study, and medical data of 189 participants was received (8.0% of those approached). One participant who completed the questionnaire online entered their Participant ID number incorrectly and could not be linked to a practice to request their notes review data. Due to the Covid-19 pandemic affecting practices' capacity to continue research, requests for patient data ceased in March 2020. Four practices did not return notes reviews.

### Descriptive data

Participant characteristics are provided in Table 2. One participant did not complete the demographic questionnaire. Most respondents were female (n = 187, 67.5%) and the mean age was 57.2 years (SD = 14.6). The sample was predominantly white (n = 273, 98.5%), married or cohabiting (n = 190, 68.6%), and in employment (n = 142, 51.3%). There was no significant difference between the mean age of respondents and non-respondents (55.4 years), *t* =

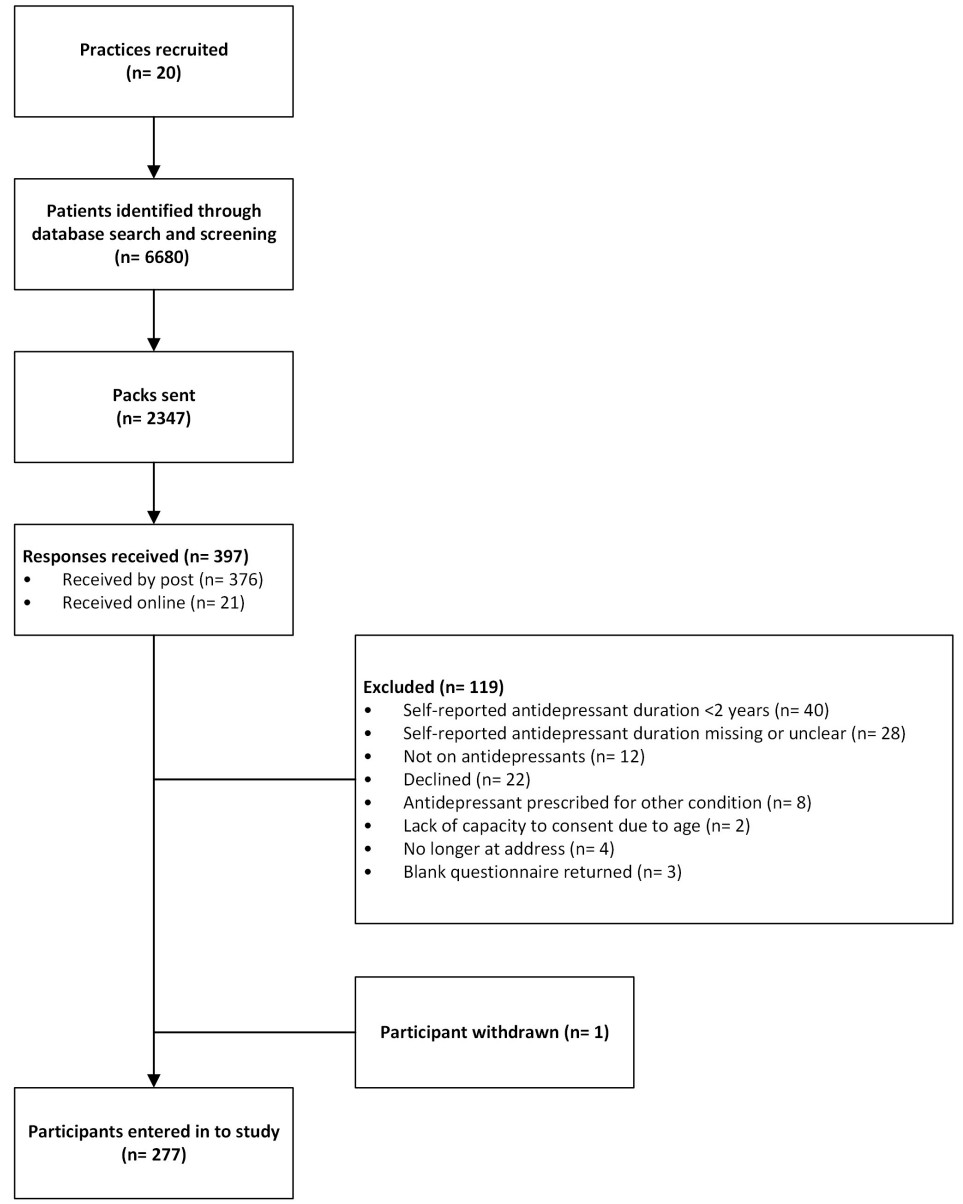

**Fig 2. Flow diagram showing participant recruitment.**

1.43, $p = 0.15$, 95% CI -0.53, 3.38; or percentage of female respondents (67.5%) and non-respondents (72.0%), $\chi^2 = 2.35$, $p = 0.13$.

Characteristics around antidepressant use and depression are presented in Table 3. The most prescribed antidepressants were Citalopram (n = 88, 31.7%) and Sertraline (n = 74, 26.7%). Participants self-reported a median current antidepressant duration of 10 years (IQR 4, 16).

The mean score for symptom severity was 8.90 (95% CI 8.06, 9.74). A higher proportion of participants reported a score of 9 or lower (n = 146, 57.3%), indicating mild to minimal depression symptom severity. More participants had attempted to stop taking antidepressants with their doctor's knowledge (n = 127, 45.8%). Ninety-two participants (33.2%) had

**Table 2. Participant characteristics.**

| Variable | | N (%) |
|---|---|---|
| Age | Mean years (SD) | 57.2 (14.6) |
| | Range | 23.3–92.5 |
| Gender | Female | 187 (67.5%) |
| | Male | 90 (32.5%) |
| Ethnicity | White | 273 (98.6%) |
| | Black Caribbean | 1 (0.4%) |
| | Asian British | 1 (0.4%) |
| | Mixed Race | 1 (0.4%) |
| | Missing | 1 (0.4%) |
| Marital status | Married/Cohabiting | 190 (68.6%) |
| | Separated/Divorced | 33 (11.9%) |
| | Widowed | 19 (6.9%) |
| | Single | 34 (12.3%) |
| | Missing | 1 (0.4%) |
| | Households with dependents at home | 120 (43.3%) |
| Education level | None | 30 (10.8%) |
| | CSE/NVQ Level 1 | 16 (5.8%) |
| | GCSE/O Level/NVQ Level 2 | 47 (17.0%) |
| | A Level/BTEC/NVQ Level 3 | 38 (13.7%) |
| | HNC/HND/City & Guilds/Teaching Qualification/NVQ Level 4 | 36 (13.0%) |
| | Degree/Higher Degree/NVQ Level 5 | 62 (22.4%) |
| | Vocational Qualification | 37 (13.4%) |
| | Missing | 11 (4.0%) |
| Work status | Employed (Full/Part time/Self-employed) | 142 (51.3%) |
| | Volunteer | 5 (1.8%) |
| | Unemployed | 6 (2.7%) |
| | Permanently Sick/Disabled | 21 (7.6%) |
| | Retired | 84 (30.3%) |
| | Homemaker | 15 (5.4%) |
| | Student | 2 (2.6%) |
| | Missing | 2 (2.6%) |

not attempted to stop taking antidepressants at all, compared to 39 participants (14.0%) who had tried to come off antidepressant both with and without their doctor's knowledge. Eighty-one participants (30%) reported successfully stopping antidepressants in the past, with 71 (87.7%) reporting a median treatment-free duration of 3 years (IQR 1.2, 10) before restarting treatment.

## Outcome data

Prescribing outcomes were recorded at six months for 175 participants (92.6% of participants with notes review data). (Table 4). The majority (n = 153, 87.4.%) did not change their antidepressants dose, compared to 16 participants (9.1%) who reduced their dose or stopped altogether.

Of the 14 participants who reduced their dose, 11 had a face-to-face appointment with their GP, and one participant had a medication review with a pharmacist. One participant did not have any appointments with a health professional, and no data were provided for the

Table 3. Antidepressant treatment history and depression symptom severity.

| Variable | | N (%) |
|---|---|---|
| Current antidepressant | Amitriptyline | 5 (1.8%) |
| | Citalopram | 88 (31.7%) |
| | Clomipramine | 2 (0.7%) |
| | Dosulepin | 1 (0.4%) |
| | Duloxetine | 6 (2.2%) |
| | Escitalopram | 5 (1.8%) |
| | Fluoxetine | 33 (11.9%) |
| | Lofepramine | 4 (1.4%) |
| | Mirtazapine | 18 (6.5%) |
| | Paroxetine | 11 (4.0%) |
| | Sertraline | 74 (26.7%) |
| | Trazadone | 1 (0.4%) |
| | Venlafaxine | 25 (9.0%) |
| | Missing | 4 (1.4%) |
| Age first prescribed antidepressants (N = 262) | Mean (SD), median (LQ, UQ) | 39.4 (15.42), 38 (27, 50) |
| Current antidepressant duration (months, N = 277) | Mean (SD), median (LQ, UQ) | 134.5 (114.52), 120.0 (48, 192) |
| Previous attempts to stop (N = 277) | With doctor's knowledge | 127 (45.8%) |
| | Without doctor's knowledge | 97 (35.0%) |
| Successful discontinuation (N = 270) | | 81 (30%) |
| Duration antidepressant-free (months, N = 71) | Median (SD) | 36.00 (123.16) |
| Current depression symptom severity (N = 255) | Mean, Median (SD) | 8.9, 8.0 (6.81) |
| | Minimal (0–4) | 83 (30.0%) |
| | Mild (5–9) | 63 (22.7%) |
| | Moderate (10–14) | 49 (17.7%) |
| | Moderately severe (15–19) | 34 (13.3%) |
| | Severe (20–27) | 26 (10.2%) |
| | Missing | 22 (7.9%) |

final participant. For the two participants that stopped completely, one had a face-to-face appointment with their GP, and the other stopped requesting antidepressant prescriptions. Fifty-two participants (29.7%) had a face-to-face appointment and eight (4.6%) had a telephone appointment with their GP. Two participants who did not change their antidepressant dose had a medication review with a pharmacist.

## Beliefs about depression and antidepressant discontinuation

Mean scores for beliefs and attitudes towards depression and antidepressant discontinuation are shown in S1 Table. The mean scores for intention (M = 2.44, 95% CI 2.23, 2.65) and subjective norms (M = 2.35, 95% CI 2.21, 2.49) towards starting to come off antidepressants were low. Higher mean scores indicated stronger beliefs that depression was *chronic* (M = 4.65, 95% CI 4.46, 4.84) and *medication* was needed to help control/cure depression (M = 5.12, 95% CI 4.96, 5.28).

Most participants strongly agreed or agreed that they were comfortable with taking antidepressants (n = 234, 84.8%), and nearly all participants (n = 248, 90.2%) strongly agreed or

Table 4. Prescribing data at six months taken from medical notes reviews.

| Outcome | N (%) |
|---|---|
| Change in prescription (N = 175) | |
| Increase | 4 (2.3%) |
| No change | 153 (87.4%) |
| Reduce | 14 (8.0%) |
| Stopped | 2 (1.1%) |
| Changed antidepressant type | 2 (1.1%) |
| Prescription request method (N = 179) | |
| Appointment | 26 (14.5%) |
| Reception | 106 (59.2%) |
| Online | 42 (23.5%) |
| Telephone | 3 (1.7%) |
| Repeat box | 2 (1.12%) |

agreed that they understood why they were prescribed antidepressants. Conversely, most participants disagreed or strongly disagreed that they were taking antidepressants they no longer needed (n = 189, 68.5%) or that their antidepressants were giving them side effects (n = 163, 59.0%). Sixty-seven (24.2%) participants indicated they were uncertain around whether they would like to stop taking their antidepressants, or their willingness to stop taking antidepressants if their doctor said it was possible (n = 100, 36.2%). However, if participants were to start to come off antidepressants, over half (n = 167, 60.3%) reported they would be comfortable if their doctor were involved with the process as well as providing follow-up compared to a nurse practitioner (n = 109, 39.4%) or pharmacist (n = 161, 58.1%). Most participants (n = 240, 87.3%) indicated a preference for face-to-face follow-up appointments with their GP.

## Global beliefs and attitudes towards discontinuation

Multiple linear regression was conducted for 173 participants (i.e., complete cases) to determine whether Global beliefs were associated with attitudes towards antidepressant discontinuation. The means, standard deviations, and intercorrelations for global beliefs on attitude are presented in S2 Table. The multiple correlation coefficient ($R = 0.71$) showed a moderate to strong linear relationship between global beliefs and attitudes towards stopping antidepressants $F(5, 167) = 33.03$, $p < 0.001$, adj. $R^2 = 0.48$.

The coefficients for each of the outcome variables are shown in S3 Table. Stronger beliefs in the necessity of antidepressants, along with stronger beliefs that depression can be cured/controlled by medication, has a physical cause, and chronic timeline were significantly associated with more negative attitudes towards stopping antidepressants. Necessity of antidepressants were most strongly associated with attitudes towards discontinuation ($B = -0.16$, 95% CI -0.21, -0.12 $p < 0.01$). However, concerns about antidepressants were not significantly associated with attitudes towards stopping antidepressant treatment ($B = 0.04$, 95% CI -0.01, 0.09, $p = 0.06$).

## Factors associated with intentions

The regression analysis for intentions was run on complete data from 161 participants. The means, standard deviations and correlations between variables are shown in S4 Table. Most variables had a significant linear relationship with intentions. Intentions were shown to have moderate to strong significant linear correlations with attitudes ($r = 0.75$, $p < 0.001$) and subjective norms ($r = 0.75$, $p < 0.001$). Necessity, medication to cure/control, and a chronic

timeline were all found to have moderate significant negative linear relationships with intention. Attitudes towards discontinuing antidepressants had moderate significant linear correlations with PBC ($r = 0.59$, $p < 0.001$) and necessity ($r = -0.61$, $p < 0.001$).

The results from each step in the multiple regression are presented in Table 5. The results showed that the three constructs from the TPB accounted for significant variation in intention scores $F(3, 157) = 118.04$, $p < 0.001$, adj. $R^2 = 0.62$. The addition of global beliefs (Step 2) to the outcome of intention led to a small but significant increase $R^2$ change of 0.04, $F(5, 152) = 3.31$, $p = 0.007$. There was a minimal change in $R^2$ when adding past history to the model (Step 3), but this change was not significant $F(3, 149) = 1.8$, $p = 0.14$. The addition of symptom severity (Step 4) and duration of antidepressant treatment (Step 5) did not change $R^2$.

Each of the models were tested to see whether they were statistically significant in the explaining intentions. The full model including all constructs from the TPB, global beliefs, past history, symptom severity and antidepressant treatment duration to explain intention was statistically significant $R^2 = 0.69$, $F(13, 147) = 24.17$, $p < 0.001$, adjusted $R^2 = 0.65$.

The regression coefficients show that attitude ($B = 0.54$, 95% CI 0.29, 0.78, $p < 0.001$), subjective norm ($B = 0.39$, 95% CI 0.13, 0.66, $p < 0.001$) and PBC ($B = 0.20$, 95% CI 0.03, 0.37, $p = 0.016$) added statistically significantly to explaining intentions. No linear relationships were found between global beliefs, symptom severity, or current duration of antidepressant treatment. Within the variable of past behaviour, previous attempts to stop taking antidepressants with a doctor's knowledge and successfully stopping showed a positive linear relationship on intentions to discontinue antidepressants, but were not statistically significant ($B = 0.37$, 95% CI -0.03, 0.72, $p = 0.06$ and $B = 0.22$, 95% CI -0.33 -0.68, $p = 0.39$ respectively). Taking all variables into account, only TPB constructs and concerns maintained their ability to explain intentions towards starting to come off antidepressants.

## Predicting behaviour

A Mann-Whitney U test was run for 151 participants to determine whether there were differences in either intention of PBC scores between those who reduced (N = 12, 7.95%) or did not reduce (N = 139, 92.05%) their antidepressants at six months. Median intention scores for participants who reduced (2.00) and did not reduce (1.67) was not statistically significantly different, $U = 1049.50$, $z = 1.52$ $p = 0.128$. There was no statistically significant difference in median PBC scores between those who reduced (3.33) and did not reduce (3.00), $U = 772.00$, $z = -0.42$, $p = 0.668$ [65].

The binomial logistic regression model was found to be non-significant, $\chi^2(1) = 0.83$, $p = 0.36$. Variation in having an appointment with a health professional or not was less than 1%. The model showed no improvement in estimating the probability of having an appointment with a health professional compared to a model that assumed that all cases would be classified as not attending an appointment. The model's sensitivity was poor in that it did not correctly predict any participants who did have an appointment (n = 60). The specificity of the model was high in that all participants (n = 105) who did not have an appointment with a health professional were correctly predicted not to have had an appointment. The odds of having an appointment increased with stronger intentions towards starting to come off antidepressants, but this finding was not statistically significant, OR = 1.09, 95% CI 0.91, 1.31, $p = 0.36$.

## Discussion

The aim of this study was to determine whether beliefs and attitudes towards long-term antidepressant use are associated with intentions to stop treatment; and whether these intentions translate into actual behaviour, using an extended model of the TPB.

**Table 5. Factors associated with intentions using TPB variables, global beliefs, past history, symptom severity and antidepressant duration.**

| | Step 1 B | β | p | 95% CI of B | Step 2 B | β | p | 95% CI of B | Step 3 B | β | p | 95% CI of B | Step 4 B | β | p | 95% CI of B | Step 5 B | β | p | 95% CI of B |
|---|---|---|---|---|---|---|---|---|---|---|---|---|---|---|---|---|---|---|---|---|
| Constant | −1.32 | | <.001 | −1.87, −0.78 | −0.55 | | .564 | −2.43, 1.33 | −1.67 | | .163 | −4.02, 0.68 | −1.73 | | .151 | −4.09, 0.64 | −1.63 | | .175 | −4.00, 0.74 |
| Attitude | 0.63 | **.50** | <.001 | 0.46, 0.80 | 0.55 | **.44** | <.001 | 0.37, 0.74 | 0.54 | **.43** | <.001 | 0.36, 0.73 | 0.54 | **.43** | <.001 | 0.36, 0.73 | 0.54 | **.43** | <.001 | 0.35, 0.72 |
| Subjective Norm | 0.45 | **.28** | <.001 | 0.26, 0.63 | 0.36 | **.22** | <.001 | 0.17, 0.54 | 0.39 | **.24** | <.001 | 0.20, 0.58 | 0.40 | **.25** | <.001 | 0.21, 0.59 | 0.39 | **.24** | <.001 | 0.20, 0.58 |
| PBC | 0.21 | **.16** | .008 | 0.06, 0.36 | 0.21 | **.16** | .009 | 0.05, 0.37 | 0.22 | **.17** | .006 | 0.06, 0.38 | 0.21 | **.17** | .011 | 0.05, 0.38 | 0.20 | **.16** | .016 | 0.04, 0.37 |
| Necessity | | | | | 0.02 | .05 | .417 | −0.03, 0.08 | 0.02 | .05 | .455 | −0.04, 0.08 | 0.03 | .05 | .411 | −0.03, 0.08 | 0.03 | .06 | .364 | −0.03, 0.09 |
| Concern | | | | | 0.06 | **.14** | .010 | 0.02, 0.11 | 0.06 | **.14** | .014 | 0.01, 0.11 | 0.07 | **.15** | .013 | 0.01, 0.12 | 0.07 | **.16** | .010 | 0.02, 0.12 |
| Physical | | | | | −0.02 | −.01 | .804 | −0.15, 0.12 | −0.01 | −.01 | .853 | −0.15, 0.12 | −0.01 | −.01 | .908 | −0.14, 0.13 | −0.01 | −.01 | .843 | −0.15, 0.12 |
| Chronic | | | | | −0.12 | −.09 | .092 | −0.26, 0.02 | −0.08 | −.07 | .251 | −0.23, 0.06 | −0.08 | −.06 | .303 | −0.22, 0.07 | −0.06 | −.05 | .402 | −0.21, 0.09 |
| Medication | | | | | −0.10 | −.07 | .212 | −0.27, 0.06 | −0.10 | −.07 | .250 | −0.26, 0.07 | −0.09 | −.06 | .265 | −0.26, 0.07 | −0.11 | −.07 | .215 | −0.027, 0.06 |
| With doctor | | | | | | | | | 0.35 | .09 | .068 | −0.03, 0.72 | 0.36 | .10 | .061 | −0.02, 0.74 | 0.37 | .10 | .054 | −0.01, 0.75 |
| Without doctor | | | | | | | | | 0.01 | .00 | .963 | −0.35, 0.37 | 0.00 | .00 | .993 | −0.36, 0.36 | −0.01 | .00 | .964 | −0.37, 0.35 |
| Successfully stopped | | | | | | | | | 0.18 | .04 | .416 | −0.25, 0.60 | 0.18 | .04 | .404 | −0.25, 0.61 | 0.22 | .06 | .312 | −0.21, 0.66 |
| Symptom severity | | | | | | | | | | | | | −0.01 | −.03 | .581 | −0.04, 0.02 | −0.01 | −.03 | .554 | −0.04, 0.02 |
| Antidepressant duration | | | | | | | | | | | | | | | | | 0.00 | −.06 | .294 | 0.00, 0.00 |
| Adjusted $R^2$ | 0.63 | | | | 0.67 | | | | 0.68 | | | | 0.68 | | | | 0.68 | | | |
| $\Delta F(df_1, df_2)$, $p$ value | $\Delta F(3, 157) = 89.10, p < .001$ | | | | $\Delta F(5, 152) = 3.31, p = .007$ | | | | $\Delta F(3, 149) = 1.84, p = .143$ | | | | $\Delta F(1, 148) = 0.31, p = .581$ | | | | $\Delta F(1, 147) = 1.11, p = .294$ | | | |

Notes: Step 1: Effects of Attitude, Subjective Norm, and Perceived Behavioural Control (PBC) on intentions towards stopping antidepressant use in the next six months. Step 2: Step 1 plus Global Beliefs. Step 3: Step 2 plus Past Behaviour. Step 4: Step 3 plus Current antidepressant duration. Step 5: Step 4 plus Symptom Severity. Significant effects are in bold.

Overall, most participants had little to no intention to start to come off antidepressants, and fewer than 10% of the sample had started to reduce their antidepressant dose at six months. The full model was found to significantly explain 65% of the variance in intentions towards starting to come off antidepressants; with more favourable attitudes and subjective norms the strongest factors associated with intentions to stop long-term antidepressant treatment.

Most participants believed their depression was chronic, felt taking antidepressants was necessary, and were generally comfortable with taking them. Necessity beliefs about antidepressants appeared to be the most important factor associated with attitudes towards starting to come off antidepressants, with over 85% of participants agreeing that taking antidepressants was necessary. Furthermore, the proportion of variance of concerns in explaining attitudes and intentions was small, in line with other research showing that patients may not prioritise concerns about taking antidepressants over the perceived risks of discontinuation [30,66–71]. Stronger beliefs in the necessity of antidepressants are related to higher levels of adherence in the initial stages of antidepressant treatment; with beliefs in the necessity of antidepressants continuing to increase over time [34,72–74]. This may explain the findings that stronger beliefs in the chronicity of depression are associated with fewer intentions to discontinue treatment.

While few participants started to come off antidepressants, an important finding is that subjective norms were significant in explaining intentions to start to come off antidepressants. While limited, there is some qualitative evidence to suggest that patients may experience pressure from significant others to consider antidepressant discontinuation[24,75]; whereas others may feel inclined to start to come off antidepressants if they feel they would be supported by family members or friends during the process of discontinuation [70,76]. Most participants stated they would be comfortable if their doctor gave them support and follow up if they were to discontinue treatment, which is supported by existing literature [30,70,76–78]. Previous attempts to stop with a doctors' knowledge showed a positive association towards intentions to stop antidepressants, and 11 out of the 16 participants who successfully started antidepressant discontinuation had a face-to-face appointment with their GP. Ongoing monitoring and review and a positive relationship with the GP is important for patients to receive appropriate guidance and support during the acute and maintenance phase of treatment and could facilitate decision-making around stopping treatment and subsequent discontinuation [76,79–81].

As 85% of prescriptions were issued using remote methods, few participants had face-to-face contact with their GP. Requesting prescriptions remotely may limit opportunities for patients to talk about their antidepressant use and potential discontinuation with their GP. Trials have shown that prompting GPs to review their patients' long-term antidepressant use will result in a proportion of patients discontinuing. In unselected samples, around 6-8% will discontinue antidepressants after practitioner review [22,82]. If the patients are selected for their willingness to try discontinuing, the rate is higher than 40%, with little risk of relapse of depression, at least up to 12 months after discontinuation [83].

Patients should be encouraged to attend more face-to-face consultations to discuss management, long-term risks of antidepressant use, and continued support, should they wish to discontinue treatment [15,29,30,76,84]. Internet and telephone support for discontinuing antidepressants, after primary care practitioner review and advice about tapering the dose, can reduce the risk of depressive and withdrawal symptoms, and conserve mental wellbeing [85]. However, both patients and GPs demonstrate uncertainty about who is responsible for initiating a consultation to review their antidepressant use [21,27,70,81,86]. Furthermore, there is limited guidance on how to initiate discussions around discontinuation or how to manage patients' fears and uncertainties, with varying levels in GP confidence when listening

to and managing patients' fears and concerns around discontinuing long-term antidepressant use [26,27,77].

## Strengths & limitations

To the best of our knowledge, no previous research has explored how beliefs and attitudes are associated with intentions towards long-term antidepressant discontinuation or examined the strength of the TPB in explaining behaviours regarding antidepressant use. The findings suggest that the utility of the TPB in explaining intentions towards discontinuing long-term antidepressant use is similar to its utility when applied to other health-related behaviours, where it has been shown to explain between 40-49% of the variance in intentions [87]. However, there are some limitations of the TPB that should be considered. While it is acceptable to add additional explanatory variables to the model, they should only be added if they can show a significant proportion of variance in intentions or behaviour in addition to the original constructs of the TPB [37]. Past behaviour is considered one of the strongest predictors of future behaviour, but only if it is performed frequently [43,48]. In the current study, past history accounted for very little change in the variance of the model in explaining intentions, but could be rationalised in that antidepressant discontinuation is not frequently performed [30,69,71]. The intention-behaviour gap should also be considered [88]. Some participants with little intention to start to come off antidepressants did eventually reduce or stop their antidepressant dose. As the majority of participants that stopped or reduced their dose had an appointment with their GP or a pharmacist, this suggests that a review consultation could act as an implementation intention and may 'bridge the gap' between behaviour and intentions [88–92].

A key limitation of the research was the level of missing data and the use of self-report data to measure current antidepressant duration. Participants self-reported a median continuous antidepressant treatment duration of 11 years, which is considerably higher than the average reported length of treatment in previous research [22,93,94]. As some participants stated they did not know how long they had been on antidepressants, this self-reported higher duration of treatment could be based on participants' best guess rather than prescribing data reported in published data. This, along with many participants responding that they did not know how long they had been taking antidepressants for, is an interesting finding. Despite testing the face validity of the questionnaire through cognitive interviews, many responses for the current study were left blank or were difficult to interpret. Furthermore, as most participants indicated little to no intention to start to come off antidepressants in the next six months, and less than 10% of participants reduced their antidepressant dose at six months, it was difficult to create a reliable model. Due to missing data,the error may be over-estimated and the power of the model is low, reducing the likelihood that a statistically significant result shows a true effect, and making it difficult to rule out a Type II error [95]. This may be the case for concerns about antidepressants and previous attempts to stop with the doctor's knowledge, as the results found for these possible explanatory variables were in the direction of a positive association and approached the 5% level of statistical significance ($p = 0.06$ in both cases). Therefore, while the models explaining intentions were more robust than the model predicting behaviour, it is not possible to make reliable inferences about how well the TPB can explain intentions to stop long-term antidepressant treatment.

The sociodemographic characteristics of participants who took part in the study should be considered. Nearly all participants were from a White ethnic group, so the findings may not represent the beliefs and attitudes of patients from ethnic minority backgrounds. Research [96] suggests that people from ethnic minority backgrounds have weaker beliefs in the biological causes of depression compared to people from a White ethnic background, and have

stronger beliefs in the psychosocial causes of depression. This may explain why people from ethnic minority backgrounds are less likely to believe that antidepressants are effective in managing depression [97], and hold stronger beliefs that antidepressants are addictive [96]. Difficulties in recruiting underrepresented groups to mental health research are unfortunately not uncommon [98]. Overall health-related deprivation patterns are evident in England, with significant health inequalities between the North and the South of the country, which can be explained by socioeconomic deprivation [99]. Participants were recruited through GP practices based in the South and South-West of England. Therefore, the participating practices may not represent those from areas with higher levels of socioeconomic deprivation. Future research needs to explore the beliefs, attitudes, and behavioural intentions towards long-term antidepressant use both between and within different sociodemographic groups.

## Implications for primary care

The research suggests a need for a more patient-centred approach to the management of depression in primary care, where the beliefs about depression and treatment preferences are key considerations when formulating a treatment plan [96,100,101]. Given strong negative views towards intentions to stop taking antidepressants and concerns around symptoms of withdrawal and relapse during discontinuation [77,81,84,102–108], further discussions between the patient and GP around beliefs and attitudes towards long-term antidepressants are needed from the outset, so patients can actively consider their intentions towards discontinuing long-term use. At the point of prescribing antidepressants, GPs should discuss the broader psychosocial determinants of depression and stress that while medication may be helpful, other approaches such as lifestyle adaptations and psychological therapies may also be of benefit. GPs could provide further information around the limited duration of antidepressant use, along with uncertainty surrounding the benefits and risks of taking antidepressants beyond two years – both in terms of iatrogenic effects, and the likelihood that individuals may still relapse whilst still on antidepressants as often as those who discontinue [1]. Better knowledge and understanding of these potential risks may then link to patients' attitudes towards starting to come off antidepressants, which in turn may be associated with stronger intentions to discontinue treatment.

Patients need to be aware of the importance of ongoing monitoring and review, so that these conversations with the GP can take place. In turn, regular monitoring and review will help maintain a strong GP-patient relationship, which could facilitate conversations around intentions to start to come off antidepressants [85]. This could give patients greater confidence to start the process of antidepressant discontinuation.

To further facilitate the process of discontinuation, GPs need appropriate guidance and support to help inform patients about the role of antidepressants in managing depression, and how to broach the conversation regarding discontinuation. In addition to informing patients at the start of antidepressant treatment that it should not be considered for life and will need to be managed slowly [109], further guidance is needed for GPs to help manage patients' fears and uncertainties about symptoms of withdrawal and relapse and appropriate guidance on the tapering process and successful antidepressant discontinuation.

Considering more global beliefs and attitudes patients may have towards the necessity of long-term antidepressants use means GPs may be able to support patients in formulating a plan for reducing their antidepressant dose that addresses their particular beliefs and mitigates any fears and uncertainties they may have.

Moreover, further research could explore patients' views about discussing antidepressant discontinuation from other health professionals, such as pharmacists or nurse prescribers.

## Supporting information

**S1 File. Study Protocol.**
(PDF)

**S2 File. Questionnaire Survey booklet.**
(PDF)

**S3 File. Medical Notes data capture form.**
(PDF)

**S1 Table. Mean scores for beliefs about depression and antidepressant discontinuation.**
(PDF)

**S2 Table. Means, standard deviations, and intercorrelations for global beliefs on attitude.**
(PDF)

**S3 Table. Global beliefs associated with attitudes towards starting to come off antidepressants.**
(PDF)

**S4 Table. Means, standard deviations and intercorrelations for beliefs and attitudes on intentions.**
(PDF)

## Author contributions

**Conceptualization:** Rachel Dewar-Haggart.

**Data curation:** Rachel Dewar-Haggart.

**Formal analysis:** Rachel Dewar-Haggart.

**Funding acquisition:** Rachel Dewar-Haggart.

**Investigation:** Rachel Dewar-Haggart.

**Methodology:** Rachel Dewar-Haggart.

**Project administration:** Rachel Dewar-Haggart.

**Supervision:** Ingrid Muller, Felicity Bishop, Adam W. A. Geraghty, Beth Stuart, Tony Kendrick.

**Writing – original draft:** Rachel Dewar-Haggart.

**Writing – review & editing:** Rachel Dewar-Haggart, Ingrid Muller, Felicity Bishop, Adam W. A. Geraghty, Beth Stuart, Tony Kendrick.

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
