## [Decision Letter · Decision Letter 0]

19 Mar 2024

PONE-D-24-05450Predicting intentions towards long-term antidepressant use in the management of people with depression in primary care: A longitudinal survey studyPLOS ONE

Dear Dr. Dewar-Haggart,

Thank you for submitting your manuscript to PLOS ONE. After careful consideration, we feel that it has merit but does not fully meet PLOS ONE’s publication criteria as it currently stands. Therefore, we invite you to submit a revised version of the manuscript that addresses the points raised during the review process.

We look forward to receiving your revised manuscript.

Kind regards,

Chi-Shin Wu

Academic Editor

PLOS ONE

Reviewers' comments:

Reviewer's Responses to Questions

**Comments to the Author**

1. Is the manuscript technically sound, and do the data support the conclusions?

Reviewer #1: Partly

Reviewer #2: Yes

2. Has the statistical analysis been performed appropriately and rigorously? 

Reviewer #1: No

Reviewer #2: Yes

3. Have the authors made all data underlying the findings in their manuscript fully available?

Reviewer #1: Yes

Reviewer #2: No

4. Is the manuscript presented in an intelligible fashion and written in standard English?

Reviewer #1: Yes

Reviewer #2: Yes

5. Review Comments to the Author

Reviewer #1: Statistical methods - Why was hierarchical regression modelling used? Why was multiple linear regression with robust SE used for some analyses and hierarchical models for others? This needs an explanation/ better write up of statistical methods.

Lines 237-238 and 247 and elsewhere - present IQR as lower quartile, upper quartile.

Table 3 - If you are going to present mean, median, (SD) present in this order mean (SD), median (LQ, UQ)

Line 269 - I would not focus on the p-value, especially as the correlation coefficient is so small. Please remove the p-value.

Lines 286-287 - along with mean and SD does not seem to fit in this sentence.

Lines 292-293 - reword to take slope coefficient out.

Line 297 - coefficient and 95%Ci would be more useful.

Is line 301 a repeat of line 290? If so, remove one.

Line 326 and elsewhere - If you are going to include p-values (would prefer it if they are not included), please give exact p-values for p-values greater than or equal to 0.001.

Table 5 needs some explaining. I assume 95%CI is for step 5. Why include robust SE beta? (again, changes to the statistical methods will help with this one). I am not sure what the column on the far right is. I would remove stars to signify statistical significance and add 95%CI for all models so that the reader can make their own judgement on the importance of each variable. Also please make it clearer what the four rows at the bottom of the table are showing.

Table 5 - check the constant for step 2.

Table 5 and elsewhere - you cannot predict an outcome from data collected at the same time. Please change the terminology to something like factors associated with. Prediction can be used for outcomes that come from the notes as they represent a later time point than the survey.

Table 6 and explanation below - It looks like you have carried out a non-parametric test, but have presented mean and (SD). If you carry out a non-parametric test, then please give median (LQ, UQ) for descriptive statistics.

Page 24, 2nd line - change expB to odds ratio.

Table S1 - please explain what the column at the far right of the table is showing. It is not clear.

Table S2 - please remove stars to represent levels of statistical significance.

Table S3 - what is the difference between the three beta coefficients in this table? If all are needed, please ensure they are accompanied by 95%CI.

Table S4 - the last row seems to have been omitted.

Reviewer #2: A very important piece of work contributing to the literature about understanding people’s long-term antidepressant medication use and factors underlying their intention to considering stopping them. The framework used to explore the determinants of intention to stop medication was appropriate and it was followed by examining actual behaviour. This could be elucidating in terms of understanding any gaps between behavioural intention and behaviour, factors that might get in the way and how to overcome them in clinical practice.

Abstract

Well-presented and clearly indicates the rationale and main findings of the study.

The abstract states that normative beliefs, and not subjective norms, were one of the stronger predictors of intention. Please clarify whether it was normative beliefs or subjective norms.

Although all TPB constructs and concerns about antidepressant use maintained their ability to predict intentions towards starting to come off antidepressants in the analysis, only “normative beliefs” and attitudes were mentioned in the abstract. Please mention all significant predictors.

Introduction

Concise and well-written. Authors clearly explain the rationale of the study.

46 – Please clarify whether what was measured was normative beliefs and not actually subjective norms.

Methods

99-100- Please justify the decision about the inclusion criterion around duration of current antidepressant medication (over two years)

121 – Please also name in the text the TPB variables that are hypothesised to predict the outcomes of interest.

123 – Error in the citation. Please correct this.

139 - Please refer to the number of items used for each direct TBP variable (attitudes, subjective norms, perceived behavioural control, intention).

144 - I would suggest rephrasing from good to acceptable as Cronbach’s α of 0.60 is not very high and many sources would say that only values over 0.70 are acceptable (e.g. Cortina, J. M. (1993). What is coefficient alpha? An examination of theory and applications. Journal of applied psychology, 78(1), 98).

145-147 - Please provide further description of the “composite variables for direct measures”. Please clarify whether the items used tapped into the belief (indirect) measures of TPB or whether they measured the direct measures directly (subjective norms, attitudes, perceived behavioural control) as this is not clear in the Methods. In TPB there are direct (subjective norms, attitudes, perceived behavioural control) and indirect measures (behavioural beliefs, normative beliefs, control beliefs) that are determinants of behavioural intention. They are not synonymous. Please provide a brief definition for attitudes, subjective norms and perceived behavioural control either as part of the Method section or in the Introduction when mentioning TPB as the framework that was used.

Please provide internal consistency estimates for each of the subscales (for each TPB determinant and intention). In case any of the items were dropped during item reliability testing, please report this.

148-149 - Please describe the constructs captured by PATD questionnaire as well as the number of items, scoring, min-max score. Please provide the internal consistency/reliability estimate for the present sample.

153 – Perhaps italicise “necessity” and “concerns” as you have done that a few lines further down for the other subscale of the measures to be consistent in the way of presenting information. Please provide the internal consistency/reliability estimate for the present sample for this measure and its subscales.

150 - In TPB terms “salient beliefs” or readily accessible beliefs refer to the specific beliefs about the specific behaviour that is being addressed, in this case starting to come off antidepressant medication in the next 6 months. These are categorised in normative, behavioural and control beliefs about the specific behaviour. In this study, it seems that salient beliefs refer to more global beliefs about antidepressant medication or depression, and not specific to the behaviour of interest within the timeframe of interest. It might be worth rephrasing this section and wherever else the constructs captured by this questionnaire are mentioned, as “salient beliefs” commonly refers to something different in studies using a TPB framework.

160 – When referring to past behaviour, please clarify what these behaviours were in the text.

161-162- Please indicate the unit of measurement for the current treatment duration (e.g. months, years, etc)

Table 1 Please correct typo – Subjective norms

169 – Please specify at what point the medical professionals at the GP surgeries reviewed participants records (e.g., at least 6 months after participation, looking through notes for the 6-month period following participation to the study)

191-193 - Please provide rationale for conducting the Pearson’ s correlation between necessity and concern around antidepressant medication.

191- Please correct typo – Pearson’s

195 – Please use lower care letter- “Salient”

204 - “Transformation of the variables” is mentioned with no previous indication to what this refers to and what analysis was conducted regarding this. Please provide clarification on the transformation of any variables in the study and the reason for doing so.

Results

245 – It is not clear what the authors meant by “successfully stopping” medication. Please clarify if that was defined using specific criteria in the questionnaire.

272-282 – It is unclear whether these findings relate to yes/no questions. For example, is it that 24.2% responded that they are certain about stopping taking their antidepressants and they authors have interpreted this as the majority being uncertain? Please clarify what these percentages refer to as it is not clear.

Discussion

Line numbering stopped at page 23.

First paragraph: The way this is presented is that TPB was used and applied to the study to help identify factors that might determine participants’ intention (and actual behaviour). The last sentence is somewhat contradictory, as it is stating that the TPB was applied after measuring participants’ beliefs and attitudes to see if it could explain them. Please rephrase for clarity.

Second paragraph: Here, normative beliefs are mentioned as a significant predictor of intention. However, in the model shown in figure 2 normative beliefs are not included. Please clarify. It is important to note that in TPB terminology “normative beliefs” (refers to indirect measure of TPB and different items are used to measure this) is a different construct to “subjective norms” (direct measure of TPB).

Third paragraph

1st line: please correct grammar error – “felt taking medication was necessary”.

Here, it stated that “necessity” was the most important predictor when considering stopping antidepressant medication. From the analysis and results, it seems that necessity was linked with attitudes towards coming off the medication however, it seems that they did not contribute significantly towards the intention of doing it. Please consider how this might be the case and be more precise in the interpretation of this result as “considering” stopping the medication seems to relate more to the intention rather than the favourable/ unfavourable attitudes towards stopping the medication.

Fourth paragraph (page 25 of the submitted manuscript): It is mentioned that subjective norms significantly predicted intention. However, the next sentence refers to results in other items and specifically, items on page 8 and item 33, which was not a subjective norm item. Although this could help further explain the significant effect of subjective norms being a significant predictor of intention, it is not the same thing.

Please discuss this finding in relation to existing relevant literature around subjective norms and their importance on predicting intention of coming off antidepressants. It is also important to discuss the difference between subjective norms that refer to a specific behaviour (coming off antidepressants in the next 6 months) in comparison to general attitudes about being comfortable to stop medication if doctors would support participants to do so at some point in the future if it was discussed. Additionally, subjective norms do not only refer to their doctor’s views about stopping medication but also to “most people important” to the participant, “people who care about “ the participant, and the social pressure more broadly. Considering the intention to stop medication was low on average, it might be worth considering how other people’s views might impact participants’ intention.

Implications for primary care: The first paragraph summarises relevant literature that builds the rationale for conducting this study, which would be more appropriate for an introduction section, rather than linking clearly specific findings from this research to clinical practice. Please refer to the implications with clear reference to the specific findings of this study (e.g., importance of necessity beliefs that are linked to attitudes, which in turn were significant predictors of intention).

Pages 6 and 7 of the APPLAUD questionnaire seem to refer to “indirect” measures of TPB (belief items) that were not mentioned in the Methods section of the manuscript, and it is unclear if they were used in the analysis and how. Only the direct measures of TPB have been included in the analysis model as presented by the authors. Please clarify.

6. PLOS authors have the option to publish the peer review history of their article (what does this mean? ). If published, this will include your full peer review and any attached files.

**Do you want your identity to be public for this peer review?** For information about this choice, including consent withdrawal, please see our Privacy Policy .

Reviewer #1: No

Reviewer #2: No

---

## [Author Response · Author response to Decision Letter 0]

9 Sep 2024

Thank you for reconsidering our manuscript. We have now responded to the reviewers' comments and have included our responses in a table, attached as a document to our resubmission. Please let us know if you require any further information.

---

## [Decision Letter · Decision Letter 1]

10 Nov 2024

PONE-D-24-05450R1Predicting intentions towards long-term antidepressant use in the management of people with depression in primary care: A longitudinal survey studyPLOS ONE

Dear Dr. Dewar-Haggart,

Thank you for submitting your manuscript to PLOS ONE. After careful consideration, we feel that it has merit but does not fully meet PLOS ONE’s publication criteria as it currently stands. Therefore, we invite you to submit a revised version of the manuscript that addresses the points raised during the review process.

We look forward to receiving your revised manuscript.

Kind regards,

Chi-Shin Wu

Academic Editor

PLOS ONE

Journal Requirements:

Reviewers' comments:

Reviewer's Responses to Questions

**Comments to the Author**

1. If the authors have adequately addressed your comments raised in a previous round of review and you feel that this manuscript is now acceptable for publication, you may indicate that here to bypass the “Comments to the Author” section, enter your conflict of interest statement in the “Confidential to Editor” section, and submit your "Accept" recommendation.

Reviewer #1: (No Response)

Reviewer #3: (No Response)

2. Is the manuscript technically sound, and do the data support the conclusions?

Reviewer #1: Yes

Reviewer #3: Yes

3. Has the statistical analysis been performed appropriately and rigorously? 

Reviewer #1: Yes

Reviewer #3: I Don't Know

4. Have the authors made all data underlying the findings in their manuscript fully available?

Reviewer #1: No

Reviewer #3: Yes

5. Is the manuscript presented in an intelligible fashion and written in standard English?

Reviewer #1: Yes

Reviewer #3: Yes

6. Review Comments to the Author

Reviewer #1: Thank you for the changes you have made so far. I have a few more statistical comments:

I used the version with the yellow highlighting to do my review so line numbers are based on that version.

Line 235 - change multivariate to multivariable. They have different meanings and you mean multivariable. https://bjo.bmj.com/content/101/10/1303

Line 240 - I am not sure that what you are describing is hierarchical regression. I think you are just adding variables sequentially to a linear regression model. If you are using hierarchical modelling, can you state what the random effect is in the methods.

Line 319 - Should 134 be 234 to make the percentage make sense? I assume the n~277.

Line 323 - is 24.2 a percentage?

Line 343 - 0.16 should be -0.16.

Line 343 - The p-value is not 0.00. Increase the number of decimal places or use p<0.01.

Reviewer #3: Thank you for the opportunity to review this important paper. This paper covers the important topic of long-term antidepressant use in primary care patients. The study is well-motivated with the relevant scientific literature, and well-conducted with an underlying theory and adequately adapted questionnaires. The study is very relevant, certainly for primary care, the site with the most antidepressant users. The study quantitatively extends knowledge from previous qualitative studies.

My comments mainly concern the Discussion.

1. Implications for primary care: In my opinion the practical value of the paper improves with making a distinction between starting an antidepressant and how to manage follow-up prescriptions. The authors have shown that stronger beliefs in the necessity of antidepressants, stronger beliefs that depression can be cured with antidepressants, that depression has a physical cause and is chronic, are all related with more negative attitudes towards discontinuation. These beliefs are certainly also induced by prescribing GPs. GPs should take care of avoiding to mention these issues but instead stress the social causes of depression, the fact that medication can be helpful but is to be considered only part of the treatment alongside lifestyle adaptations and conversations about the context of depression. Moreover, GPs should always stress the limited duration of antidepressant use. For the follow-up prescriptions, GPs should be clear about scheduled face-to-face consultations and so on (as the authors already describe in the paper.)

2. May be this is a personal preference, but I would like a separate description of the main results of the study and the comparison with the literature. Now, these paragraphs seem to overlap.

A few minor comments:

1. The study title indicates that this is a longitudinal study while in the Methods (page 6 line 105 the authors use the term cross-sectional.

2. I’m a bit confused about the numbers: (a) on page 14 line 268 there are 189 participants of whom medical data were received, while on page 19 line 298 prescribing outcomes were obtained for 175 participants and on page 20 line 332 multiple linear regression was performed for 173 participants. This is probably due to missing responses. Maybe a short statement about this is sufficient. (b) in Figure 2 the numbers are not correct: in the box with excluded participants, the numbers for the different reasons for exclusion do not add up to 119.

7. PLOS authors have the option to publish the peer review history of their article (what does this mean? ). If published, this will include your full peer review and any attached files.

**Do you want your identity to be public for this peer review?** For information about this choice, including consent withdrawal, please see our Privacy Policy .

Reviewer #1: No

Reviewer #3: No

---

## [Author Response · Author response to Decision Letter 1]

4 Dec 2024

Thank you for your comments and suggestions. We have responded to all your comments and these are included in the 'Response to reviewers' document attached to this revised submission. We hope our responses are acceptable.

Line 235 - change multivariate to multivariable. They have different meanings and you mean multivariable. https://bjo.bmj.com/content/101/10/1303

Thank you for highlighting this error and for taking the time to include a link to the BJO paper. We have changed the term multivariate to multivariable (Line 235).

Line 240 - I am not sure that what you are describing is hierarchical regression. I think you are just adding variables sequentially to a linear regression model. If you are using hierarchical modelling, can you state what the random effect is in the methods.

Thank you for this comment. For our study, the term ‘hierarchical regression’ was used to explain that variables were added in a sequential fashion, but prioritised according to the theoretical importance of each variable. We appreciate that this is confusing and have referred back to Tabachnick & Fidell’s work that states that ‘sequential regression’ is sometimes referred to ‘hierarchical regression’. To avoid confusion, we have changed ‘hierarchical’ to ‘sequential’ (Line 240).

Line 319 - Should 134 be 234 to make the percentage make sense? I assume the n~277.

Thank you for highlighting this typo. We have double-checked the data and it is 234 participants and have made the correction (Line 319).

Line 323 - is 24.2 a percentage?

Thank you for spotting this. Yes, this is a percentage – we have added the missing % (Line 323).

Line 343 - 0.16 should be -0.16. Thank you again for spotting this.

We have added a negative symbol. (Line 343)

Line 343 - The p-value is not 0.00. Increase the number of decimal places or use p<0.01.

As the p-value would need to be reported to 10 decimal places (p= 0.000000000179…) we hope that it is acceptable to report the p value as <0.01 (Line 343).

Thank you for the opportunity to review this important paper. This paper covers the important topic of long-term antidepressant use in primary care patients. The study is well-motivated with the relevant scientific literature, and well-conducted with an underlying theory and adequately adapted questionnaires. The study is very relevant, certainly for primary care, the site with the most antidepressant users. The study quantitatively extends knowledge from previous qualitative studies.

We would like to thank the reviewer for their very kind comments about our paper.

1. Implications for primary care: In my opinion the practical value of the paper improves with making a distinction between starting an antidepressant and how to manage follow-up prescriptions. The authors have shown that stronger beliefs in the necessity of antidepressants, stronger beliefs that depression can be cured with antidepressants, that depression has a physical cause and is chronic, are all related with more negative attitudes towards discontinuation. These beliefs are certainly also induced by prescribing GPs. GPs should take care of avoiding to mention these issues but instead stress the social causes of depression, the fact that medication can be helpful but is to be considered only part of the treatment alongside lifestyle adaptations and conversations about the context of depression. Moreover, GPs should always stress the limited duration of antidepressant use. For the follow-up prescriptions, GPs should be clear about scheduled face-to-face consultations and so on (as the authors already describe in the paper.)

Thank you for your comments. We have added a sentence (Lines 516 – 519) to emphasise the need to discuss psychosocial causes of depression along with lifestyle adaptations at the point of prescribing antidepressants. We have also added ‘the limitation duration of antidepressant use’ (Line 520) to highlight your second point. We hope this strengthens the implications section of the discussion. These implications are explored (and explained) further by the qualitative findings from this mixed-methods study, and RDH is preparing the manuscript for publication in due course.

2. May be this is a personal preference, but I would like a separate description of the main results of the study and the comparison with the literature. Now, these paragraphs seem to overlap.

Thank you for this comment – we have considered structuring the discussion in the way you suggested, but feel that this could extend the length of the manuscript, which is already relatively long. We will therefore leave the discussion section in its current structure; however we welcome any decision from the editors regarding this.

1. The study title indicates that this is a longitudinal study while in the Methods (page 6 line 105 the authors use the term cross-sectional.

We agree that this is confusing for the reader – we have changed Line 105 to define the study as ‘longitudinal’.

2. I’m a bit confused about the numbers:

(a) on page 14 line 268 there are 189 participants of whom medical data were received, while on page 19 line 298 prescribing outcomes were obtained for 175 participants and on page 20 line 332 multiple linear regression was performed for 173 participants. This is probably due to missing responses. Maybe a short statement about this is sufficient.

Thank you for this observation and we understand that this may be confusing for the reader. We do mention in the ‘Statistical methods’ section that data analysis was conducted using complete cases. We have added “(i.e. complete cases)” to Line 333 to clarify why n=173. We also acknowledge a key limitation of the study was the level of missing data, despite testing the face validity of the questionnaire prior to its implementation in the current study (Lines 477-484).

We contacted all 20 practices to request notes review data for all participants who were entered into the trial (n= 277) – we received notes review data for 189 participants, with the reasons for the outstanding 88 participants mentioned in Lines 268-272. We have amended the statement in Line 298-299 to say that prescribing outcomes were recorded at 6 months for 175 participants (92.6% of participants with notes review data) for clarity.

(b) in Figure 2 the numbers are not correct: in the box with excluded participants, the numbers for the different reasons for exclusion do not add up to 119

Thank you for highlighting this – when converting the table to a TIF file, the last two exclusion criteria (participant no longer at address and blank questionnaire returned) were cut from the image. The TIF image has been corrected.

---

## [Editor Report · Decision Letter 2]

18 Dec 2024

Predicting intentions towards long-term antidepressant use in the management of people with depression in primary care: A longitudinal survey study

PONE-D-24-05450R2

Dear Dr. Dewar-Haggart,

We’re pleased to inform you that your manuscript has been judged scientifically suitable for publication and will be formally accepted for publication once it meets all outstanding technical requirements.

Kind regards,

Chi-Shin Wu

Academic Editor

PLOS ONE
---

## [Editor Report · Acceptance letter]

PONE-D-24-05450R2

PLOS ONE

Dear Dr. Dewar-Haggart,

I'm pleased to inform you that your manuscript has been deemed suitable for publication in PLOS ONE. Congratulations! Your manuscript is now being handed over to our production team.

Kind regards,

on behalf of

Dr. Chi-Shin Wu

Academic Editor

PLOS ONE